# Therapeutic Metabolic Reprograming Using microRNAs: From Cancer to HIV Infection

**DOI:** 10.3390/genes13020273

**Published:** 2022-01-29

**Authors:** Mark S. Gibson, Cláudia Noronha-Estima, Margarida Gama-Carvalho

**Affiliations:** BioISI—Biosystems & Integrative Sciences Institute, Faculty of Sciences, University of Lisboa, 1749-016 Lisboa, Portugal; msgibson@fc.ul.pt (M.S.G.); cfestima@fc.ul.pt (C.N.-E.)

**Keywords:** microRNAs, metabolism, immunometabolism, therapy, cancer, HIV

## Abstract

MicroRNAs (miRNAs) are crucial regulators of cellular processes, including metabolism. Attempts to use miRNAs as therapeutic agents are being explored in several areas, including the control of cancer progression. Recent evidence suggests fine tuning miRNA activity to reprogram tumor cell metabolism has enormous potential as an alternative treatment option. Indeed, cancer growth is known to be linked to profound metabolic changes. Likewise, the emerging field of immunometabolism is leading to a refined understanding of how immune cell proliferation and function is governed by glucose homeostasis. Different immune cell types are now known to have unique metabolic signatures that switch in response to a changing environment. T-cell subsets exhibit distinct metabolic profiles which underlie their alternative differentiation and phenotypic functions. Recent evidence shows that the susceptibility of CD4^+^ T-cells to HIV infection is intimately linked to their metabolic activity, with many of the metabolic features of HIV-1-infected cells resembling those found in tumor cells. In this review, we discuss the use of miRNA modulation to achieve metabolic reprogramming for cancer therapy and explore the idea that the same approach may serve as an effective mechanism to restrict HIV replication and eliminate infected cells.

## 1. Energy Metabolism as a microRNA-Regulated Process

Twenty years ago, a set of landmark papers established microRNAs (miRNAs) as a numerous and highly conserved class of functional small non-coding RNAs across all eukaryotes [1,2,3]. Currently, there are 576 bona fide miRNA genes annotated in the human genome [4], although the actual number may be higher [4,5], with a significant proportion exhibiting non-ubiquitous, cell type-specific expression patterns [6]. These genes range from independent, single or clustered transcriptional units to intron-contained, protein coding gene-associated genomic loci. Expression of miRNA genes relies on canonical and non-canonical processing pathways to generate 21–23 nt long molecules that complex with Argonaute proteins to form the RNA Induced Silencing Complex (RISC) [7]. The key common feature of all these genes is their ability to generate a transcript containing a ~70 nt long stem-loop that is recognized by the Dicer protein at the final step of their biogenesis pathway, although some rare cases of Dicer-independent miRNAs have been reported [7,8].

Following the first descriptions of a functional role for miRNAs in *C. elegans* development [9,10] it is now clear that these molecules can influence every aspect of cellular function and even function in long-distance, cell-to-cell regulation [11]. Notwithstanding its absolute centrality to life processes, energy metabolism is no exception to this rule.

### 1.1. An Overview of Energy Metabolism

Humans (and other chemotrophs) derive their energy from the oxidation of fuel molecules, such as glucose, amino acids, glycerol and fatty acids. The final aim of the process is the formation of ATP, the universal currency of free energy in biological systems. Depending on the starting point, ATP generation will engage different albeit convergent enzymatic pathways [12]. Indeed, following the hydrolysis of foodstuffs into their smaller units most fuel molecules will be converted into acetyl-CoA, with a small amount of ATP being generated. This carrier brings acetyl units into the citric or tri-carboxylic acid (TCA) cycle for complete oxidation into CO_2_. Glycolysis, the conversion of glucose into pyruvate, is the predominant and nearly universal pathway for the generation of metabolic energy in biological systems. The oxidative decarboxylation of pyruvate into acetyl-CoA establishes the link between this reaction and the TCA cycle.

For each oxidized acetyl group that enters the TCA cycle, four electrons are transferred to the carrier co-enzymes NAD^+^ and FAD, generating their reduced NADH and FADH2 equivalents. The flow of electrons from these carriers to oxygen in the mitochondrial respiratory chain supports the regeneration of NAD^+^ and FAD while constituting the most efficient process for ATP production in cell metabolism. Indeed, while the conversion of glucose to acetyl-CoA generates two ATP molecules, the metabolism of this carrier through the TCA cycle and the oxidative phosphorylation (OxPhos) process i.e., the synthesis of ATP from the phosphorylation of ADP using energy obtained through the electron transport chain, will lead to an estimated net yield of 30 ATP molecules [12].

In addition to glucose, the degradation of other fuel molecules supports the direct production of ATP molecules and generates intermediates for the TCA cycle. For example, conversion of the amino acid glutamine or any of the other five carbon (C5) amino acids to glutamate, followed by de-amination by glutamate dehydrogenase, allows them to enter the TCA cycle as α-ketoglutarate (α-KG). Lipid-derived fatty acids can be used as a source of energy through their oxidation into acetyl- or succinyl-CoA, through which they enter the TCA cycle. The energy efficiency of the fatty acid β-oxidation process is similar to glycolysis, generating significant amounts of NADH and FADH2 that transfer their electrons to the respiratory chain. However, because the entry of acetyl-CoA into the TCA cycle relies on its conjugation with oxaloacetate, which is a product of carbohydrate metabolism, these two processes are interdependent [12].

### 1.2. Proliferating Cells Favor Aerobic Glycolysis and Glutaminolysis

Most cells in the human organism rely on cellular respiration to generate the ATP required for homeostasis. In the absence of oxygen, the anaerobic regeneration of NAD^+^ and FAD can be achieved through the conversion of pyruvate into lactate, shunting this intermediate away from the TCA cycle. The process of anaerobic glycolysis is typical of contracting muscle, a setting when the rate of ATP consumption exceeds both the oxygen supply to the respiratory chain and the speed at which the TCA cycle can use pyruvate. However, it has been known for a long time that highly proliferating cells will favor glycolysis over the OxPhos process, even in the presence of oxygen [13]. This is justified by two different aspects. On the one hand, although less efficient, aerobic glycolysis occurs at a faster pace than OxPhos, which is dependent on the number of mitochondria in the cell, something that cannot be modified as quickly as the availability of glycolytic enzymes. On the other hand, and more importantly, glucose is a major provider of intermediates for the biosynthetic processes that need to take place in order for cells to be able to divide. Indeed, glucose is the major carbon source for nucleotide and lipid biosynthesis and for the synthesis of several amino acids [13]. These biosynthetic reactions require reducing, rather than oxidative power, which is provided by NADPH, the cellular electron donor workhorse. NADPH is generated by the pentose phosphate pathway (PPP) when the glycolytic intermediate glucose-6-phosphate is oxidized to ribose-5-phospate. In turn, this five-carbon sugar is the building block of critical components of metabolic pathways such as ATP, CoA, NAD^+^, FAD, as well as being a nucleotide precursor.

Another relevant source for the generation of NADPH is the conversion of malate into pyruvate by the malic enzyme [12]. Glutamine, the most abundant amino acid in the blood, has been shown to be a major source for this process [14]. In addition to this, glutaminolysis, the conversion of glutamine to lactate, named after the similarities it has with glycolysis, provides a very significant contribution to bioenergetics and macromolecular synthesis in proliferating cells. Glutaminolysis has the ability to generate more ATP molecules than glycolysis, and can supply 30 to 50% of the ATP requirement of proliferating cells [13].

### 1.3. Energy Metabolism Is Tightly Linked to Biosynthesis

As mentioned above, in addition to energy, proliferating cells require a vast array of metabolites, from nucleotides to lipids, in order to undergo replication and cell division. This implies a tight coupling between energy metabolism and biosynthesis to ensure the availability of all required intermediates. Many of the carbon and amine precursors required for biosynthetic processes are derived from TCA cycle intermediates. Indeed, while in non-proliferating cells the main role of the TCA cycle seems to be to maximize ATP production through the complete oxidation of substrates to CO_2_, in proliferating cells this cycle becomes a critical provider of intermediates for biosynthesis. These intermediates need to be replenished in case their abundance reaches critical levels, with the direct conversion of pyruvate into oxaloacetate being a core anaplerotic (from the Greek for “to fill up”) reaction in mammals [12]. In proliferating cells, glutamine is a major contributor to anaplerosis, as its carbons are efficiently converted into acetyl-CoA [13]. It is a major precursor for amino acid synthesis, and the main amino group donor for nucleotide synthesis.

Notwithstanding the potential use of glutamine as an energy source, the use of glucose is so central to energy metabolism that it can be synthesized from non-carbohydrate precursors, namely lactate and some amino acids, including glutamine. This occurs through a process called gluconeogenesis. Pyruvate represents the entry point into the gluconeogenesis pathway for lactate and C3 amino acids, whereas C4 amino acids enter it through conversion to oxaloacetate. Oxaloacetate is an intermediate of the TCA cycle and can be used by cells as a precursor in amino acid biosynthesis along with another intermediate, α-KG. This connection further illustrates the degree to which energy metabolism and biosynthetic processes are tightly linked, requiring very precise orchestration.

Despite the relevance of gluconeogenesis, most of the glucose used in metabolism is derived from the degradation of glycogen, a branched polymer of glucose residues that serves as a readily mobilizable fuel store in the liver. The processes of glycogen synthesis and degradation are tightly coupled to the sensing of circulating glucose levels. In fact, as exemplified above, all energy and biosynthetic pathways are interlinked and coordinated by nutrient, intermediate, and energy sensing systems that establish different kinds of metabolic regulation.

### 1.4. Metabolism Is Highly Regulated by Signalling Pathways and miRNAs

From all that has been discussed above, it becomes clear that energy metabolism and biosynthesis are highly integrated and dynamic processes that must adapt quickly and robustly to both extrinsic and intrinsic factors. In order to achieve this, different levels of regulatory mechanisms have to be in place. Regulatory events control physiological processes such as food ingestion and the release of stored molecules into circulation, thereby determining the availability of fuel molecules to cells. These regulatory events define the predominant reactions that support the generation of acetyl-CoA. The insulin and glucagon signaling pathways are central in determining the availability of glucose as the primary fuel for ATP generation. Depending on the availability of fuel molecules and the respective need for energy or biosynthesis, regulatory processes that directly adjust the production and efficiency of metabolic enzymes and their pathways are put in place and/or coordinated through specific signaling pathways. These include the mechanistic target of rapamycin (mTOR) and hypoxia-inducible factor 1-α (HIF-1α) signaling. The mTOR pathway coordinates anabolic and catabolic processes with the availability of nutrients, energy, and oxygen as well as with growth factor signaling [15]. Signaling through HIF-1α, the transcription factor Myc and phosphatidylinositol-3-kinase (PI3K) are critical promoters of metabolism in proliferating cells, acting to stimulate aerobic glycolysis and glutamine metabolism [16].

All of these regulatory processes can involve the control and coordination of protein abundance, the core function of miRNAs. Accordingly, miRNAs have been shown to influence multiple aspects of metabolism, either through direct control of the expression levels of metabolic enzymes or by influencing regulatory signaling pathways. Specific miRNAs regulating glucose and glutamine metabolism, gluconeogenesis, the TCA cycle, the pentose phosphate pathway, lipid metabolism, insulin and glucagon signaling have been characterized in a multitude of studies. These results have been summarized in several recent (and not so recent) reviews [17,18,19,20], and will thus only be specifically mentioned in the context of the following sections. Figure 1 presents a summary of some of the most relevant microRNA-dependent regulatory events to be considered in the context of the regulation of glucose metabolism between quiescent and proliferating cells.

## 2. Metabolic Adaptations of Cancer Cells

Cancer cells are characterized, among other things, by relatively high levels of uncontrolled proliferation of clonal origin. While for solid cancers, this proliferation initially occurs in situ, sooner or later a cancer cell will acquire a set of mutations that support the invasion of adjacent tissue and even entry into the circulatory or the lymphatic system. From there, the cancer cells can be transported to a secondary location where the process of tumor growth starts over. These events of localized cellular multiplication will eventually lead to a large mass of cells—the tumor—that may reach a volume whereby diffusion of gases and nutrients becomes limited, with cells having to sustain their proliferation under hypoxic conditions. To overcome this situation, cancer cells produce growth factors to induce vascularization by normal cells, leading to more favorable growth conditions for the tumor (reviewed in [21]). Tumors are thus part of a complex local environment containing both proliferating cancer cells and normal cell types, including multiple classes of immune cells that can recognize and attack the tumor. All these cells can be “manipulated” by the tumor cells to support (or at least not interfere with) their growth.

### 2.1. Cancer Cell Metabolic Strategies Sustain High Proliferation in Nutrient and Oxygen-Poor Environments

As highly proliferating cells, cancer cells need to adopt the metabolic strategies discussed in Section 1. In fact, aerobic glycolysis was initially identified almost 100 years ago as a property of tumors by Otto Warburg and colleagues, when they observed that cancer cells use up much larger quantities of glucose than normal cells, and excrete huge amounts of lactate, independently of oxygen availability [13]. Although this observation led Warburg to propose a link between respiratory chain defects and cancer, as discussed in the previous section it is now clear that aerobic glycolysis is a general feature of highly proliferating cells and can even occur in association with enhanced mitochondrial activity [13]. A similar increase in the use of glutamine by cancer cells has been reported in several studies, along with the enhancement of the TCA cycle, PPP, lipid catabolism and other biosynthetic processes frequently found in highly proliferating normal cells (reviewed in [21]). The main difference between cancer and proliferating normal cells in this regard is that, in the latter, the uptake of fuel molecules is strictly coordinated by growth factor signaling [22]. In contrast, cancer cells constitutively scavenge glucose and glutamine from the extracellular medium thanks to mutations in critical metabolic regulatory genes such as PI3K, protein kinase B (Akt), Ras and c-myc [21]. Changes in the activity of these and other genes result in cell-autonomous increases in the abundance and activity of membrane transporters and metabolic enzymes that support enhanced glycolysis and glutaminolysis in the absence of extracellular signaling. This allows for an efficient coupling of energy and biosynthetic requirements for uncontrolled proliferation. Of note, depending on the specific pathways underlying tumor transformation different metabolic profiles can be established [23], and may even lead to cancer cell “addiction” to a given fuel molecule. For example, Wise and colleagues reported that oncogenic Myc activation selectively induced addiction to glutamine [24]. The availability of extracellular metabolites in the tumor microenvironment (TME) can also support tumor cell growth and survival. This may occur when pro-tumorigenic molecules are not degraded, or following catabolism by tumor cells, thus creating a nutrient poor environment which impairs the function of nearby immune cells. Well-known examples include lactate, amino acids, fatty acids and nucleotides [25]. The presence of certain nucleotide phosphates in the TME may actually promote anti-tumor immune responses by increasing the presence and activities of antigen-presenting cells and T cells to augment tumor cell lysis [25,26].

As mentioned above, given that cell proliferation eventually results in local nutrient and oxygen scarcity, tumor cells engage in opportunistic modes of nutrient acquisition, including capture of extracellular macromolecules which are then subjected to lysosomal degradation and release into the intracellular environment [21]. These adaptive metabolic pathways can be pushed to the point of self-catabolism through macro-autophagy, enabling cancer cells to survive through long periods of external nutrient deprivation. This process, which is found in normal cells as well [22], seems to be critical in cancer cells for maintenance of the mitochondrial activity required for sustained tumor growth in vivo [27,28].

### 2.2. LDHA Is a Critical Player in Tumor Metabolic Reprograming and Suppression of Immunosurveillance

Hypoxic conditions in the local tumor environment lead to the expression of HIF-1α, a key regulator of metabolism introduced in Section 1.4. HIF-1α signaling coordinates a cellular response program supporting both metabolic reprograming to cope with the lack of oxygen and promotion of angiogenesis to improve O_2_ delivery (reviewed by [13,29]). Among other things, HIF-1α activation leads to increased expression of glucose transporters, glycolytic enzymes, and of a central player in aerobic glycolysis, lactate dehydrogenase (LDHA), which is responsible for converting pyruvate into lactate. HIF-1α further contributes to aerobic glycolysis by blocking the conversion of pyruvate to acetyl-CoA by pyruvate dehydrogenase through the up-regulation of its inhibitory kinase, PDK [30]. This results in a decreased flux of pyruvate into the TCA cycle, promoting its conversion into lactate. Of note, many oncogenic driver mutations, such as those activating mTOR, PI3K or AKT signaling, induce HIF-1α expression even in the absence of hypoxic conditions, thus favoring the switch to a metabolic state favorable for high levels of cellular proliferation [13,29]. HIF-1α metabolic reprograming occurs in tight cooperation with the Myc transcription factor [31], which promotes glutaminolysis and mitochondrial biogenesis. Additionally, Myc is a direct activator of the transcription of glucose transporters and glycolytic enzymes [32], as well as of LDHA [33]. The relevance of LDHA for tumor development and growth is underscored by the fact that its inhibition by shRNA or small drug-like molecules can reduce tumor cell growth both in vitro and in vivo [34,35]. These observations led to the proposal that therapeutic targeting of energy metabolism could provide an effective new avenue for controlling cancer. Notwithstanding, Boudreau and colleagues recently reported the presence of innate and acquired resistance to LDHA inhibition linked to tumor metabolic plasticity, namely through the activation of the MPK–mTOR–S6K signaling pathway leading to increased OxPhos [36].

The impact of LDHA inhibition on tumor growth in vivo seems to extend beyond a cell-autonomous effect and to involve the enhancement of immunosurveillance mechanisms. Infiltrating immune cells, including macrophages, natural killer (NK) cells, dendritic cells and CD4^+^ and CD8^+^ T lymphocytes, have the ability to mount an immune response in the TME that can ultimately control proliferating cancer cells [37]. However, the proliferative and metabolic reprogramming that tumor cells engage in has profound impacts on the TME. Indeed, the enhanced capture of nutrients, reduction of oxygen levels, and secretion of high amounts of lactate make for a hostile, acidic environment that severely limits the ability of the immune system to mount an effective response [38]. This seems to be largely dependent on the level of LDHA activity, as shown by studies comparing the growth of tumors from control melanoma or LDHA-shRNA knock-down (Ldha^low^) cells [39]. Despite similar levels of proliferation in vitro, mice injected with Ldha^low^ cells developed smaller tumors in strict dependence on an active immune system and correlating with the amount of tumor-infiltrating cells. This study shows that the accumulation of lactic acid inhibits the function and survival of T lymphocytes and NK cells, contributing to the establishment of immunosuppression in the TME. Furthermore, high lactate levels in the TME have been shown to induce M2-like polarization of tumor-infiltrating macrophages, an immunosuppressive phenotype [40].

These and other studies (recently reviewed by [38]) reveal the degree to which tumor metabolism can interact with normal and in particular immune cells in the TME, and influence their own metabolism in ways that act to promote cancer progression. In fact, a recent study highlights how metabolic stress induced by the TME in conjunction with immunological signaling from persistent antigen stimulation can drive T cell exhaustion in cancer [41]. These new observations add strength to the concept of therapeutic metabolic reprograming as an effective tool in the control of tumor progression.

## 3. miRNAs in the Therapeutic Modulation of Cancer Metabolism

It is widely acknowledged that miRNA regulatory networks are dysregulated in cancer, contributing to its development. This is the case for tumor suppressive and oncogenic miRNAs (oncomiRs) as well as for key proteins required for miRNA biogenesis such as Ago2, Dicer and Drosha (reviewed in [42]). Typically, tumor suppressive miRNAs are downregulated and oncomiRs upregulated in cancerous cells. Hence, restoring miRNAs of interest to their endogenous levels is an obvious approach to treatment. Modulating miRNA levels in order to engineer desirable therapeutic outcomes has therefore been a long-standing objective [43]. Currently, one of the most cutting-edge methods to treat diseases involves targeted delivery of mimics or anti-miRNA molecules (antagomiRs) to directly restore or suppress selected miRNAs. Despite the formidable challenges of delivering such therapeutics in vivo while ensuring optimal efficacy and specificity, several promising candidates have already entered clinical trials (reviewed in [42,44]). Table 1 summarizes finished and ongoing trials using miRNAs in the context of cancer, highlighting associations with metabolic regulation.

### 3.1. miRNAs Regulating Metabolic Pathways in Cancer

As discussed in Section 1, multiple components of metabolic pathways are validated miRNA targets including several proteins, the expression of which is determinant for cancer metabolic reprograming (reviewed in detail in [18,51]).

Glucose metabolism emerges as one of the critical points for miRNA regulation in cancer, with several studies identifying miRNAs that can act to either promote aerobic glycolysis or inhibit it. For example, transfecting breast cancer cell lines with miR-143 mimics significantly reduced the rate of glucose consumption through direct targeting of hexokinase [52], with the reverse observed following miR-143 inhibition. These findings were reciprocated using murine tail vein xenografts involving miR-143 mimic-containing breast cancer cell lines. In these mice, glucose uptake, lactate production, tumor volume and metastasis were all repressed [52]. In contrast, it is well established that oncogenic miR-155 upregulates glycolysis by repressing miR-143, a negative regulator of hexokinase [52] (Figure 1). This seems to occur through the modulation of the PIK3R1-PDK/AKT FOXO3a-cMYC axis, thus positively impacting on other key regulatory points for tumor energy metabolism [53]. Of note, miR-155 is known to regulate other aspects of metabolism, namely lipid biosynthesis (reviewed by [17]) and modulation of insulin signaling through a paracrine mechanism, thus influencing overall glucose homeostasis [54].

As discussed in Section 2, LDHA inhibition prevents the conversion of pyruvate into lactate, inhibiting the process of aerobic glycolysis, which can be critical for tumor growth. Several miRNAs have been shown to target the mRNA encoding this protein and to be negatively correlated with LDHA levels in human colorectal cancer samples [55]. Moreover, the same study showed that overexpression of these miRNAs in colorectal cancer cell lines was able to suppress tumor growth by inhibiting glycolysis. Among the LDHA-regulating miRNAs is the miR-34/449 family, composed of six different miRNAs sharing the same seed sequence [56]. The miR-34 miRNAs were shown in a seminal paper to be part of the p53 regulatory network and to function as tumor suppressor miRNAs [57]. Accordingly, this family of miRNAs has been shown to be downregulated in a number of different types of cancer (reviewed in [58]). Transcriptional activation of the miR-34 genes is driven by p53 during the response to DNA damage, and many of the mRNA targets they regulate are involved in cell cycling, apoptosis, metastasis and oncogenesis [59]. As the miR-34 family can target LDHA directly, it was proposed that its role in tumor suppression could involve restricting glycolysis [58]. Indeed, a handful of studies have provided support for this link for miR-34a-5p in vitro and in vivo [55,60]. In addition to regulating LDHA, a recent study has shown that miR-34c-5p is a negative regulator of the glycolytic enzyme fructose-bisphosphate aldolase A [61]. Interestingly, this miRNA establishes a regulatory loop with a long non-coding RNA that can control tumor growth by promoting aerobic glycolysis through the sponging of miR-34c-5p. Of note, the miR-34 family also seems to engage in the regulation of other aspects of metabolism, including mitochondrial dysfunction in metabolic syndrome [62].

miRNAs can interfere with tumor metabolism through key signaling pathways. For example, transfection of non-small cell lung cancer (NSCLC) cell lines with miR-31-5p mimics was shown to lead to an increase in cell proliferation, glucose uptake and the production of GLUT 1, GAPDH and LDHA proteins [63]. The opposite outcome was achieved with specific miR-31-5p inhibition. These effects were apparent using miR-31-5p mimics in mice, which promoted tumor growth. The underlying mechanism involves disruption of the regulation of glycolytic enzymes by HIF-1α, as miR-31-5p directly targets the HIF-1α inhibitor [63].

Interestingly, some oncogenic DNA viruses have been shown to express miRNAs that interfere with metabolic pathways. The Kaposi’s Sarcoma Herpesvirus (KSHV) and the Epstein Barr Virus (EBV) account for a considerable percentage of the DNA virus-associated cancers. Although their oncogenic mechanisms have been strongly linked to the NF-κB signaling pathway [64], the modulation of cellular metabolism by viral miRNAs has been clearly shown to promote a metabolic shift akin to the Warburg effect. KSHV-encoded miRNA expression reduces mitochondrial biogenesis and oxygen consumption while increasing lactate secretion and glucose uptake and promoting aerobic respiration through the activation of HIF-1α signaling [65]. The EBV-encoded microRNA EBV-miR-BART1 has been shown to be highly expressed in many EBV-associated tumors. Work by Ye and colleagues [66] has shown that this induces modifications in cellular metabolism by changing the glycolytic flux through phosphoglycerate dehydrogenase (PHGDH), which has previously been shown to contribute to oncogenesis [67]. These examples emphasize the power of miRNAs to contribute to the metabolic reprograming required for cancer cells to proliferate efficiently. 

### 3.2. Therapeutic Modulation of Metabolism in Cancer: The Power of miRNAs

The multiple observations showing the ability of miRNAs to inhibit tumor growth in different experimental models of cancer have advocated for therapeutic metabolic reprograming through the development of miRNA mimics or antagomiRs as novel avenues for the treatment of cancer.

miR-34a-5p was one of the first to be explored in this context, with early results proposing mimic replacement therapy as an attractive option for treating cancer [68]. In 2014, a phase I clinical trial was established to determine the efficacy of a synthetic, double-stranded miR-34a mimic encapsulated in a liposomal nanoparticle in adult patients with solid tumors that would not respond to traditional treatment. The trial results were disappointing, with only 4% of patients exhibiting a partial response (PR) and 24% having stable disease (SD), and had to be closed early due to serious immune-mediated adverse effects [45]. Although it remains unclear whether these effects were related to the manipulation of miR-34a-dependent networks or to non-specific inflammatory effects related to the mimic formulation, the authors highlighted the value of proof-of-concept for dose-dependent modulation of relevant target genes in the context of cancer therapy [45].

Much more promisingly, a later study established an acceptable safety profile for the treatment of recurrent malignant pleural mesothelioma (MM) using an miR-16 mimic, with one of twenty-two patients showing an objective response that lasted for 32 weeks and 68% displaying SD [46]. In this innovative approach, direct delivery of the mimics to tumor cells was attempted through the use of nonliving bacterial minicells (nanoparticles) termed TargomiRs targeting the Endothelial Growth Factor Receptor (EGFR), which is frequently overexpressed in this tumor. The rationale behind this therapy was the observation that miR-16 family members are often down-regulated in MM, acting as tumor suppressors. In a prior in vitro study, restoration of miR-16 levels was shown to inhibit tumor cell proliferation, linked to the downregulation of Bcl-2 and CCND1 [69]. In the context of the clinical trial, the authors were unable to confirm the in vivo delivery of the miRNAs to the tumor site or to detect changes in Bcl-2 and CCND1 expression. It is, however, interesting to note that two recent articles have implicated miR-16 in the regulation of glycolysis in tumors of differing origin, suggesting that metabolic reprograming could be an alternative mode of action [70,71].

Another promising study investigated the effect of cobomarsen, an LNA-modified oligonucleotide anti-miR-155 inhibitor. Recently published results have reported its effects on Diffuse Large B-cell Lymphoma (DLBCL) cell lines and a xenograft mouse model, including increased apoptosis, reduced cell proliferation and lower tumor volume. Data from a single patient receiving cobomarsen revealed a reduction in tumor size, stable disease, and the absence of toxicity [47]. Though selected miR-155 targets were examined, none of them are directly linked to metabolism. However, as discussed before, miR-155 is known to augment glucose metabolism by repressing miR-143, suggesting that metabolic reprograming may contribute to the observed effects [52]. The publicly available study design (I.D: NCT02580552) specifies a 66-patient cohort, suggesting a comprehensive report on the impact of this therapeutic awaits release.

The outcomes of these studies make a strong case for the power of therapeutic miRNA modulation in tumor cells, possibly through unexplored effects on metabolism. They also bring a number of considerations to the fore. Clearly, success in animal models targeting a single miRNA does not necessarily translate to human subjects. In addition to sharing many of the challenges of targeted therapies in cancer including tumor and individual heterogeneity, specificity, and delivery issues, the complexity of microRNA regulatory networks remains very much uncharted territory. The fact that miRNAs are natural coordinators of cellular processes linked to a given phenotype may represent a significant benefit for their use as therapeutic tools; or, on the contrary, the fact that they interfere with multiple pathways may represent a limitation due to their pleiotropic effects. This concept prompts consideration of how reprograming a single metabolic pathway (e.g., glycolysis) impacts the capacity of tumors to continue to thrive. Current thinking dictates a simultaneous increase in both glycolysis and OxPhos occurs during tumorigenesis, superseding the simpler idea of a switch from OxPhos to glycolysis [72]. Therefore, unravelling the underlying biology of how these pathways may be affected by a single miRNA mimic or antagonist could be vital in refining the treatment model. As mentioned, miRNAs such as miR-34a-5p have been shown to influence both glycolysis and mitochondrial activity. Finally, the fact that most miRNAs act as fine-tuners rather than regulatory switches can be a pro or a con. The use of metabolic flux analysis in “intact” tumors in order to better establish where to make interventions with metabolic inhibitors in clinical trials has been proposed [72]; this experimental approach could be appropriated to test a single miRNA or combinations of miRNAs, and may shed light on the success or failure of past clinical trials.

### 3.3. miRNA Control of the TME and Immunosupression

As mentioned in Section 2, tumor cells have a significant relationship with the adjacent cellular milieu. Selected studies have shown some of these interactions can be controlled by miRNAs that affect metabolism, or vice versa. For example, cancer-associated fibroblasts (CAFs) initiate and drive tumor formation and progression through interactions with cancer cells. In CAFs, isocitrate dehydrogenase 3 complex (IDH3α) is downregulated by miR-424, reducing α-KG levels, which ultimately promotes glycolysis. CAFs in the TME have low levels of IDH3α, while IDH3α overexpression inhibits the differentiation of “healthy” fibroblasts into CAFs [73]. More recently, it was shown that specific oncogenes in cancer cells define the microRNome of secreted extracellular vesicles (EVs) [74]. When released into the TME, these miRNAs have the capacity to negatively regulate adjacent cells, which may include reprograming metabolic pathways to mediate a functional switch that assists tumor growth. Some of the miRNAs released in these EVs are known modulators of metabolism [74].

In addition to CAFs, the metabolism of tumor-infiltrating immune cells is affected by the TME. A timely review [75] that discusses the power of modulating metabolic pathways for immunotherapy provides a long list of clinical trials that use metabolic agents combined with immune checkpoint inhibitors to treat cancer. Conspicuously, there is not a single reference to the possibility of using miRNA therapeutics in this context, perhaps due to the nascent connection between these two subjects in the literature. Despite this, two of the clinical trials reported the use of metformin to tackle NSCLC and melanoma. While metformin is known to target mitochondrial complex I, inhibiting respiration, glucose uptake, and consequently tumor growth [76], it modulates the expression of a large number of miRNAs as well (reviewed by [77]). Many of these miRNAs target glucose metabolism, and the alteration in their levels caused by metformin has been cited as partially responsible for its tumor suppressive activity. Interestingly, Zhao and colleagues have demonstrated that the effects of the TME on CD8^+^ T cells, linked to glucose deprivation and consequent restriction of aerobic glycolysis, are mediated by altered expression of miR-101 and miR-26a in these lymphocytes [78]. Elevated levels of these miRNAs limit expression of the EZH2 methyltransferase, affecting CD8^+^ T cell function and survival.

These studies highlight how important metabolic cooperativity is likely to be in the TME, further suggesting that the modulation of miRNA expression in normal TME cells warrants further investigation. We believe it is possible that a combinatorial strategy using chemotherapeutic agents and miRNA mimics or antagomiRs to induce metabolic reprograming of both cancer and immune system cells could open up novel approaches in the context of anti-tumor immunotherapy trials.

## 4. Metabolism and microRNAs in the Immune System

The immune system, in particular tumor infiltrating immune cells, are central elements in the control of tumor cell growth. As discussed in the previous sections, the ability of these cells to mount an anti-tumor attack is highly influenced by the TME, which has a profound impact on their activity, specifically interfering with cell metabolism. In the last decade, a growing amount of evidence has revealed how metabolism contributes to immune cell activation, differentiation, and cell fate. Immunometabolism is the field of research that connects immunology with metabolism [79]. The metabolism of immune cells is mainly governed by regulatory changes in six interconnected metabolic pathways, namely, glycolysis, the TCA cycle, the PPP, fatty acid oxidation, fatty acid synthesis (FAS), and amino acid metabolism (reviewed in [80]), all of which are partly orchestrated by microRNAs.

### 4.1. Quick Primer on T Cell Immunometabolism

T cells are one of the core components of the immune response, with the majority being classified as either CD4^+^ or CD8^+^ T cells. These can belong to one of three main functional subtypes: naïve, effector, and memory cells. Upon stimulation by antigen presenting cells, naïve T cells begin to proliferate and differentiate into one of multiple effector and memory T cell subsets, each of which has a different function and a characteristic metabolic phenotype [81]. During T cell differentiation, the energy requirements vary according to the needs and functions of the specific cell subset. For example, unstimulated naïve T cells have a quiescent phenotype and are essentially metabolically inactive when compared to activated cells. When T cell receptor (TCR) stimulation occurs, T cells boost their metabolism by shifting it from OxPhos to glycolysis [82] through the upregulation of genes associated with the glycolytic pathway, including glucose transporters [83]. These changes in metabolism linked to T cell activation share significant similarities with tumor metabolic reprograming, and occur to achieve the required energetic and biosynthetic demands of the expansion stage of the immune response. As such, modulation of T cell metabolism can have a deep impact on immune responses to disease.

The processes of T cell activation, proliferation, differentiation, and apoptosis have been shown to be fundamentally regulated by the expression of a core set of miRNAs, termed “immuno-miRs” [84]. Although changes in the metabolic status of T cells are implicated in these processes, only a limited number of recent studies have begun to demonstrate the importance of miRNAs in T cell metabolism. As discussed before, miRNAs control the expression of multiple target genes associated with metabolism. Immunometabolic regulation by miRNAs has been shown in several types of immune cells including macrophages, DCs, and B and T lymphocytes (reviewed by [85]). Interference with glycolysis and key metabolic regulators such as mTOR, 5’ adenosine monophosphate-activated protein kinase (AMPK), Myc and HIF-1α appear to be the most common routes targeted by miRNAs to modulate T cell metabolism [85]. In turn, miRNAs are induced in response to T cell activation processes. One such example is the induction of miR-150 by surface CD46 co-stimulation on CD4^+^ T cells to decrease glucose through direct targeting of glucose transporter 1 (GLUT1) [86]. miR-143, initially described as an inhibitor of glucose metabolism in cancer cells [87], was recently described as regulating memory T cell differentiation through metabolic reprograming [88]. Although these studies indicate miRNAs as modulators of immunometabolism, further studies need to be conducted in order to clarify whether miRNA-mediated metabolic reprograming of T cells can be a suitable therapeutic tool in the context of diseases that involve dysfunctional cellular metabolism. Among these is AIDS (Acquired Immuno-Deficiency Syndrome), caused by the HIV viruses.

### 4.2. The Susceptibility of CD4 T Cells to HIV Infection Is Strongly Linked to Their Metabolism 

The Human Immunodeficiency Virus (HIV) was first reported in the early 1980′s [89,90] and identified as the viral agent responsible for causing AIDS, which continues to represent a worldwide healthcare concern [91]. As a member of the lentivirus genus, HIV belongs to the family of *Retroviridae*, which are characterized by having their genetic material in the form of single-stranded RNA. Several years after the discovery of HIV-1, the highly similar HIV type 2 (HIV-2) was identified [92]. HIV-2 is characterized by lower infectivity and reduced probability of progression to AIDS [93]; for this reason most studies focus on HIV-1-associated pathology.

HIV-1 infection is initiated when a virion, a complete particle that is capable of infection, encounters a cell that presents surface receptors that allow its entry. The primary cell receptor that mediates the entry of this virus is the CD4 molecule, along with co-receptors CXCR4 and CCR5 [94]. Consequently, CD4^+^ T lymphocytes are the main target cells of HIV-1 infection. Yet, not all CD4^+^ cells are equally susceptible to entry. This means that even if the immune cell presents surface receptors that permit the entry of HIV-1, there are other factors that influence the infection of a cell. The level of activation and differentiation of CD4^+^ T cells, which are intrinsically regulated by cellular metabolism, has a critical impact on cell susceptibility to HIV-1 infection [95,96]. As with all viruses, HIV-1 does not possess metabolic machinery. As such, these infectious entities hijack host cell metabolism in order to favor their replication and ensure a productive and long-lasting infection [97]. The connection between HIV-1 and the metabolic status of its host cells is increasingly clear, and is being broadly explored in the literature. It is currently known that HIV-1 selectively infects CD4^+^ T cells with high metabolic activity regardless of their activation or differentiation phenotype [98]. In particular, HIV-1 infection is favored in cells with increased mitochondrial biomass [99], glycolysis, and OxPhos [98]. For this reason, naïve T cells, which have the lowest metabolic activity of all subsets, are the least permissive to HIV infection [100].

Research on the connection between immunometabolism and HIV-1 infection has allowed the identification of several key pathways and regulators. In 2012, a pivotal work by Loisel-Meyer et al. identified GLUT1 as a regulator of HIV-1 infection [101]. This study showed that GLUT1 expression determines cell susceptibility to HIV-1 infection, as its upregulation allows the increase in intracellular glucose levels to provide energy for the early replication steps of HIV-1 [101]. The glutamine transporter SLC1A5/ASCT2 was found to influence HIV-1 replication as well [99]. Although glucose and glutamine metabolism have been known as key regulators of HIV-1 susceptibility and infection for several years, the exact contribution of each of these molecules remained unclear until recently. In 2019, it became apparent that mitochondrial metabolic pathways have a bigger impact on HIV-1 replication than aerobic glycolysis [99]. This new work showed that glutamine fuels mitochondrial-related metabolic processes such as the TCA cycle and OxPhos, providing optimal HIV-1 infection in activated naïve and memory CD4^+^ T cells. Moreover, the exogenous addition of a TCA cycle intermediate, α-KG, was found to promote HIV-1 reverse transcription. Interestingly, the inhibition of LDHA increases HIV-1 replication, shifting glucose metabolism to the mitochondria instead of the cytoplasm [99]. α-KG can activate mTOR, which in turn regulates multiple essential cellular pathways (reviewed in [15]). mTOR signaling was shown to promote cell susceptibility to HIV-1 [102] as well as several steps of HIV-1 replication, including viral entry and transcription of HIV-1 genes [103]. Recently, mTOR was shown to control the expansion of metabolite pools, fueling viral reverse transcription, as well as nuclear import. Conversely, mTOR inhibition suppressed HIV-1 infection [102].

### 4.3. Host Cell Metabolism Influences Virion Production, Infectivity, and Latent Virus Reactivation

Glycolysis has recently been demonstrated to impact the molecular composition and functional quality of the virions produced by HIV-1 infected cells [104]. To understand the impact of aerobic glycolysis on HIV-1 virions, cells were cultured and infected in galactose-containing medium instead of glucose in order to avoid aerobic glycolysis. The results showed that the virions produced in galactose-containing medium displayed lower infectivity than those produced in the presence of glucose due to decreased efficiency in viral entry and reverse transcription. The decreased efficiency in these two processes was caused by an inappropriate packaging of HIV-1 envelope proteins and reverse transcriptase enzyme into the newly produced virions, emphasizing that HIV-1 needs aerobic glycolysis for efficient infection [104]. These findings support previous observations that identified glycolysis as an essential metabolic pathway for virion production [105].

Other biosynthetic pathways are critical for HIV-1 replication. The PPP was shown to be a crucial regulator of HIV-1 infection, as the efficiency of HIV-1 reverse transcriptase relies on the abundance of intracellular dNTPs [106]. This is so critical for HIV-1 replication that depleting the dNTP pool through the cellular protein SAM domain and HD domain-containing protein 1 (SAMHD1) blocks HIV-1 replication in T cells [107]. This protein functions as a major cellular restriction factor in immune cells that exhibit low susceptibility to HIV-1 infection, such as macrophages and DCs. FAS seems to be crucial in the late steps of HIV-1 replication, such as viral particle assembly, release and maturation, which depend on the binding of Gag viral protein to myristic acid [108]. Another essential lipid is cholesterol, which participates in HIV-1 membrane formation as well as in the entry and egress of viral particles through lipid rafts [109]. In addition to the impact of FAS on the structure of HIV-1 virions, it seems to be required for efficient HIV-1 transcription through histone acetylation, promoting viral gene expression [110]. Interestingly, several studies indicate that HIV-1 can modulate metabolism to further optimize the cellular conditions for productive infection (reviewed in [111,112]). It is known that HIV-1 infection increases glucose uptake [113] and upregulates glucose metabolism [114], although the inherent mechanisms remain unclear. A recent study demonstrated that two mitochondria-localized proteins (NLRX1 and FASTKD5) mediate OxPhos upregulation during HIV-1 infection of CD4^+^ T cells, promoting viral replication [115], thus uncovering a novel and interesting regulatory axis for therapeutic targeting against HIV-1 infection. 

HIV-infected cells undergo either programmed cell death or become quiescent, establishing latent viral reservoirs. This transition to a latent infection is accompanied by downregulated glycolysis [116], which can be reverted by viral reactivation. Latently infected cells rely on the PPP and the antioxidant thioredoxin and glutathione systems for survival. Pro-oxidant drugs working downstream of the PPP blocked the critical role of NADPH in maintaining latency, inducing a “shock and kill”-like effect [116]. Metabolic pathways further regulate the reactivation of the latent virus. For example, the use of 2-deoxyglucose, a glycolysis inhibitor, blocked HIV-1 replication, eliminated HIV-1-infected cells, and avoided HIV-1 spread upon activation of CD4^+^ T cells from HIV-1-infected individuals under antiretroviral therapy [98]. These insights reveal potential therapeutic opportunities to control latently infected cells, the key problem that must be solved in order to cure HIV infection.

## 5. Control of HIV-1 Replication and Latency by “Metabolic” miRNAs: A Potential Therapeutic Approach?

As discussed in the previous section, HIV-1 exploits cellular metabolism to fulfil the energetic demands required for viral replication and productive infection. The blockage of key metabolic pathways has been shown to interfere with HIV-1 infection, reactivation of latent virus, and clear latently infected cells. Thus, cellular metabolic reprograming should be considered when developing novel therapies to battle HIV-1. The possibility of using miRNA manipulation as a directed strategy to achieve this aim opens up an exciting avenue for future research.

### 5.1. microRNAs Are Critical Regulators of HIV Infection

HIV can effectively enter, become integrated, and replicate in host cells via a number of defined steps. Each step requires fundamental host processes which are regulated by miRNAs. The interplay between host miRNAs and HIV has been reviewed in depth elsewhere [117]. Cellular miRNAs can either inhibit or promote both primary infection [118] and latency [119,120] through the regulation of a broad spectrum of mRNA targets, both viral and host [121,122,123]. Unexpectedly, cellular miRNAs have been shown to interact directly with the viral Gag protein, interfering with the assembly of viral particles [124]. As discussed in the previous section, CD4^+^ T cell activation provides HIV with favorable metabolic conditions for productive infection [98] and is accompanied by a significant alteration in host miRNA levels. These have been proposed to render the cells more susceptible to viral entry and replication by liberating viral RNA [125] and cellular co-factors [126] from miRNA restriction. Conversely, several miRNAs elevated in resting primary CD4^+^ T cells may contribute to maintain latency [121].

Predicting the actual impact of changing miRNA levels on cell function in general and in HIV replication in particular has proven challenging because individual target mRNAs can encode proteins that promote or restrict viral infection. Furthermore, the differences in miRNA expression in primary cells versus cell lines raise significant questions regarding the relevance of certain experimental observations to the in vivo and therapeutic context. In our previous work, we identified miR-34c-5p as a novel immunomiR induced in naïve CD4 T cells in response to TCR stimulation which was able to influence HIV replication [118]. Overexpression of miR-34c-5p was found to slow cell growth and to increase HIV replication in the Jurkat T cell line [118]. This result was relatively unexpected given that this miRNA targets the expression of the transcriptional regulator PCAF, which is known to be required for efficient HIV-1 infection in the same cell line model [123]. Interestingly, miR-34c-5p is known to target the LDHA mRNA as well [55]; thus, it is striking that when it is overexpressed, or when a specific LDHA inhibitor [99] is used, the same net outcome of increased HIV replication is observed. In another recent study, HIV-infected Jurkat cells transfected with a miR-150-5p mimic demonstrated increased GLUT 1 expression and glucose uptake and reduced HIV-mediated apoptosis [127]. Although a previous work showed that miR-150-5p overexpression reduced GLUT 1 protein in uninfected Jurkat cells [86], a clear link between the manipulation of miRNA levels and alterations in both metabolic and HIV-related parameters was reported [127].

### 5.2. microRNA-Dependent Regulation of Metabolic Pathways in the Context of Viral Infection

Although the link between miRNA regulation and HIV infection is well-established, to the best of our knowledge there are no examples in the literature that explore miRNA-mediated changes in glucose and/or glutamine metabolism in order to connect the two. Despite this, studies have shown the modulation of metabolic pathway components by miRNAs that affect the replication of other viruses. In rapid succession, two recent studies revealed a role for miR-122 in facilitating hepatitis C virus (HCV) replication [128] and lipid metabolism in vivo [129]. Through miR-122 antagonism, it was revealed that reducing cholesterol and lipid co-factors has no direct impact on HCV RNA abundance [130]. However, there is an indirect effect, as these lipid components are essential for producing and releasing virus particles. More recently, miR-122 has emerged as a regulator of glucose and glutamine metabolism, targeting pyruvate kinase M2 (PKM2) [131] and glutaminase (GLS) [132] in the respective pathways. A second miRNA, miR-27a, has been simultaneously linked to HCV replication and lipid metabolism [133]. In contrast to miR-122, miR-27a restricts HCV replication while having the same effect on lipid synthesis. Through inhibition of transcription factors needed for lipid metabolism, miR-27a limits synthesis of the components required for viral particle production. A further study in hepatoma cells has shown that miR-124 overexpression inhibits fatty acid and triglyceride metabolism to inhibit HCV production [134]. The miRNA exerted this effect by inhibiting the expression of a single carboxylester hydrolase, arylacetamide deacetylase. There are multiple other examples in the literature of miRNA level manipulation affecting viral infection. In A549 cells, twenty-one different miRNA mimics were tested for their ability to reduce viral titer following infection with different strains of Influenza A virus and respiratory syncytial virus [135]. Several mimics significantly inhibited viral particle release. While the mechanism was proposed to involve suppression of the p38 MAP Kinase pathway, seven of these miRNAs (miR-27b, miR-103/miR-107, miR-124, miR-155, miR-199a, miR-223) are known to target metabolic pathway enzymes.

### 5.3. Interactions between miRNAs and Metabolic Pathways Relevant for HIV Replication

The metabolic similarities between the tumor microenvironment and the sites of HIV-1 replication have recently been emphasized [112]. A series of solid propositions for targeting immunometabolism are laid out in the same source, although miRNA modulation is not considered. As discussed in Section 3, tumor-suppressive miRNAs that inhibit metabolic pathway enzymes are already in clinical trials to treat cancer. We propose that these same principles could be applied to reprogram T cell metabolism in order to restrict HIV. Significant recent advances have provided a refined understanding of which major metabolic pathways provide the essential support to facilitate HIV replication in CD4^+^ T cells [98,99,115]. Blocking glycolysis has been shown to suppress viral replication using metabolic inhibitors or 2-deoxy-glucose, while targeting OxPhos in mitochondria has emerged as a vital strategy in defeating HIV. Glutaminolysis and the inhibition of LDHA to redirect glucose-derived carbons into the TCA cycle elevates HIV-1 reverse transcription [99]. The specific interaction between the mitochondrial proteins NLRX1 and FASTDK5 simultaneously facilitates OxPhos and HIV replication in CD4^+^ T cells [115].

In Figure 2, we have summarized the potentially relevant pathways connecting the “metabolic miRNAs” which affect glucose and glutamine metabolism to HIV infection.

Our previous data have already shown that overexpressing miR-34c-5p, a known regulator of LDHA, increases HIV replication [118]. Its similarity to the mechanism of LDHA inhibition described above [99] is striking. Four other miRNAs (miR-34a, miR-369-3p, miR-374a and miR-4524a/b) are known to degrade LDHA [19]. One of these, miR-34a, has been shown to promote HIV-1 replication in Jurkat cells [136]. Several other miRNAs have been reported to repress key nodes in the glycolytic pathway. In addition to miR-150, miR-93, miR-133, miR-223, miR-195-3p, and miR-144 all negatively regulate GLUT 1 [19]. Glucose uptake through this receptor is a rate-limiting step for productive HIV infection in CD4^+^ T cells [101]. Of these, miR-93 is known to restrict HIV-1 replication in monocytes [137], while miR-150-5p and miR-223 have the same effect in CD4^+^ T cells [126], although the mechanism of action is proposed to involve targeting non-metabolic mRNAs. In contrast, HIV-1 transgene expression in rats is promoted by miR-144 [138]. The conversion of glucose to glucose-6-phosphate is catalyzed by hexokinase (HK2), which is directly repressed by miR-143 and miR-199a and indirectly modulated by miR-155-5p inhibition of miR-143 transcription [19]. Tumor volume is reduced by inhibition of glycolysis by miR-143 overexpression, identifying it as a potential target to restrict HIV replication. In macrophages, miR-155-5p has an anti-HIV-1 effect, again proposed to occur via non-metabolic mRNA regulation [139]. Finally, in the glycolysis pathway, PKM2 is degraded by miR-124, miR-133b, and miR-122 [19]. Although miR-133b limits HIV replication through binding to Env RNA [140], miR-124 overexpression enhances virion release from Jurkat cells [136]. Glutamine metabolism and the TCA cycle harbor a number of critical enzymes under miRNA control. These include GLS (miR-23a/b), isocitrate dehydrogenase (miR-181a and miR-183), and pyruvate dehydrogenase kinase 1 (let 7). Furthermore, miR-181c and miR-210 target complexes within the mitochondrial electron transport chain. Both miR-181a and miR-181c target SAMHD1 mRNA, a well-known HIV restriction factor, in Jurkat cells [122]. Two independent studies [141,142] have found that overexpression of miR-181a increases HIV replication in astrocytes via SAMDH1 degradation.

Taken together, we believe the individual or combined effect of modulating miRNAs at nodes in the glycolysis and glutaminolysis pathways to influence HIV-1 remains an unexplored and exciting opportunity for new therapeutic developments. In tackling this, special consideration should be given to the distinct metabolic signatures [143] and likely disparate miRNA expression profiles of naïve and differentiated human CD4^+^ T cell subsets.

## 6. Final Remarks

In this review, we have summarized the major metabolic pathways used to fuel eukaryotic cells and the signaling molecules and pathways that feed into them, highlighting the points where they are known to be under miRNA regulation. We have explained how these pathways are altered during tumorigenesis, and what effect that has on infiltrating immune cells in the tumor microenvironment. We discuss how miRNA mimics have been successfully used in vitro to restrict cancer growth through targeting glycolysis and offer our perspective on how this approach could be tailored to treat tumors in vivo. There are a number of clinical trials either completed or ongoing that utilize miRNA therapeutics to treat cancer. However, the underlying mechanisms of action are usually discussed without regard to the effects of these agents on components of the metabolic pathways. This is one of the critical points we have sought to emphasize: the possibility that metabolic reprograming is a significant aspect of their functionality that remains largely under the radar. We move on to consider the applicability of this concept in regulating HIV-1 replication and achieving control of latently infected cells. We highlight known miRNA modulators of T cell immunometabolism, an emerging area of research, and anticipate further insights yet to materialize in this area in the upcoming years. Over the past decade, a wealth of studies has evaluated the relationship between HIV and metabolism. We review recent mechanistic insights into the metabolic pathway dependence of HIV, which brings into focus a set of potential novel targets for therapeutic intervention. We propose that in parallel with current thinking regarding tumor growth, these and other key nodes can be exploited therapeutically by modulating miRNA levels to reprogram T cell metabolism in order to restrict HIV. In spite of all the unknowns, we expect that the connections between metabolism, miRNAs and therapeutic approaches to multiple diseases will be a hot topic of research in the coming years.

## Figures and Tables

**Figure 1 genes-13-00273-f001:**
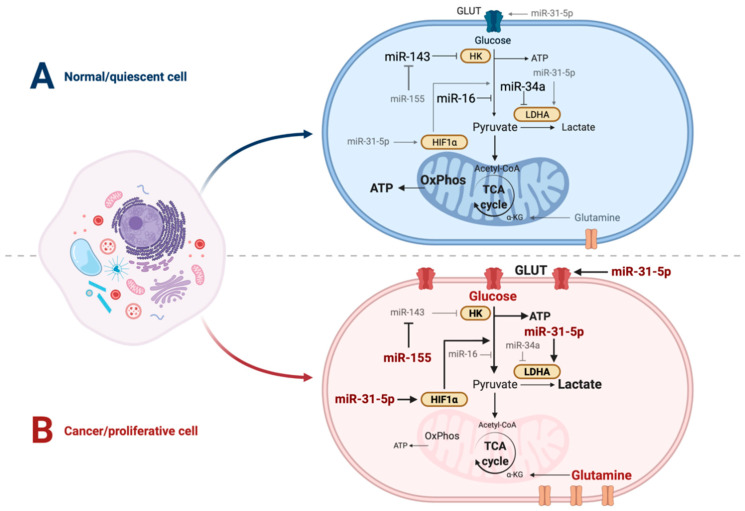
Regulation of glucose metabolism by miRNAs in quiescent versus proliferative cells. The figure depicts the main alternative energy metabolism pathways in human cells, with their relative usage in normal quiescent cells (**A**) and cancer and highly proliferative cells (**B**) represented by different levels of transparency. Known regulatory miRNAs, their direct mRNA targets (HK, LDHA and HIF1α), and certain indirect miRNA regulatory interactions with other miRNAs or pathway reactions are represented, with letter size distinguishing their abundance. Pointing arrowheads represent stimulatory interactions, while blocking arrows represent inhibitory interactions. Pathway and metabolite labels highlighted in bold and thickened lines denote their predominance in each setting.

**Figure 2 genes-13-00273-f002:**
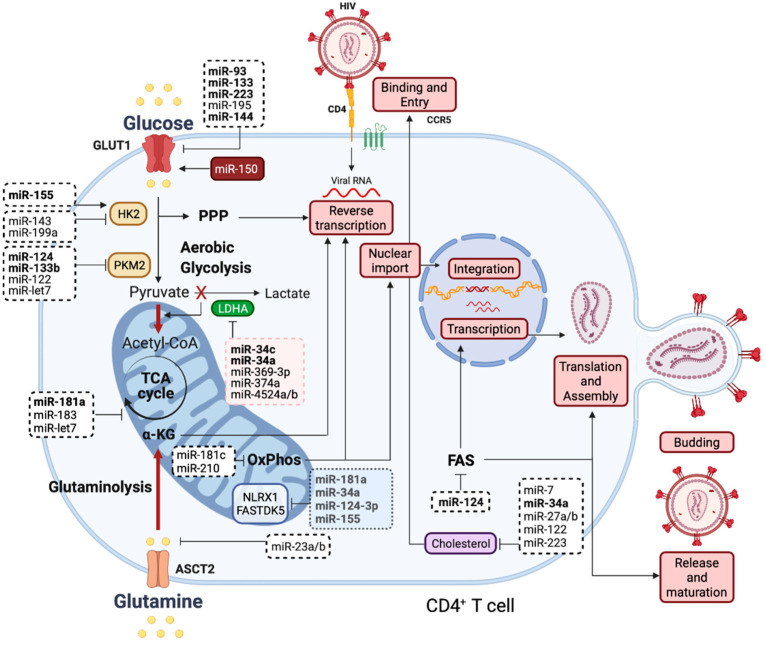
MicroRNAs and their targets within key metabolic pathways for HIV infection. The figure outlines the two main metabolic pathways that provide the nutrients required for optimal HIV-1 infection and replication in activated CD4^+^ T cells. Several enzymes and steps in these pathways are directly regulated by the indicated miRNAs. miRNAs listed in the blue box are predicted to target NLRX1 and FASTDK5 enzymes, recently shown to play a critical role in HIV infection (see main text). For simplicity, only miR-3p strands are indicated; all others are -5p. Only miR-155 (arrow) promotes the expression of its target; the rest are inhibitory. Let-7 refers to several family members. Bold miRNAs have reported effects on HIV-1 infection, as discussed in the text. miR-150 is shown to promote HIV replication through GLUT-1 (highlighted in red). Thick red lines highlight metabolic steps that promote HIV replication. Conversion of pyruvate into lactate by LDHA works against this (highlighted in green). We propose that inhibition of LDHA by miR-34 may contribute to this role in promoting HIV replication (highlighted in pink).

**Table 1 genes-13-00273-t001:** Anti-cancer miRNA therapeutics in clinical trials.

Name	Company	miRNA (form)	Target Disease	Target mRNAs Regulating Metabolism	Clinical Trial I.D.	Reference
MRX34	Synlogic	miR-34a (mimic)	Various solid tumors	LDHA (direct)	NCT01829971	[45]
MesomiR−1	EnGeneIC	miR-16 (mimic)	Malignant mesothelioma	PKM2 via ALDH1A3; PGK1 (direct)	NCT02369198	[46]
Cobomarsen	Viridian	miR-155 (antagomiR)	Diffuse Large B-cell Lymphoma	HK2 via miR-143	NCT02580552	[47]
RGLS5579	Regulus	miR-10b (antagomiR)	Glioblastoma	None reported	preclinical	[48]
INT 1B3	InteRNA Technologies	miR-193a-3p (mimic)	Solid tumors	None reported	NCT04675996	[49,50]

## Data Availability

No new data were created or analyzed in this study. Data sharing is not applicable to this article.

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
