# Peer review of "Therapeutic Metabolic Reprograming Using microRNAs: From Cancer to HIV Infection"

_genes, 2022, doi:10.3390/genes13020273_

Round 1

Reviewer 1 Report

Thanks to the authors for this very interesting manuscript. It is a very informative and well organized short review, elaborating the metabolic reprogramming by microRNAs in different contexts from cancer to HIV-1 infection. The topic is clearly of interest and relevant for research on pre-clinical and clinical level.

As suggestion, I would like the authors to insert a new paragraph on metabolites released into the tumor microenvironment and on effects these may have on immune cell reprogramming. As reference the authors could use the article PMID: 34604232

Author Response

Reviewer #1 comments

Thanks to the authors for this very interesting manuscript. It is a very informative and well organized short review, elaborating the metabolic reprogramming by microRNAs in different contexts from cancer to HIV-1 infection. The topic is clearly of interest and relevant for research on pre-clinical and clinical level. As suggestion, I would like the authors to insert a new paragraph on metabolites released into the tumor microenvironment and on effects these may have on immune cell reprogramming. As reference the authors could use the article PMID: 34604232.

Author's Reply to the Review Report (Reviewer 1)

We thank reviewer #1 for the time they took to assess our manuscript. As per the suggestion made by the reviewer, please find below some additional text to address this point. The new text makes reference to the recommended article and has been inserted into section 2.1 of the manuscript (bottom of page 6):

“The availability of extracellular metabolites in the TME can also support tumor cell growth and survival. This may occur when pro-tumorigenic molecules are not degraded or following catabolism by tumor cells, thus creating a nutrient poor environment, which impairs the function of nearby immune cells. Well-known examples include lactate, amino acids, fatty acids and nucleotides [25]. The presence of certain nucleotide phosphates in the TME may actually promote anti-tumor immune responses by increasing the presence and activities of antigen presenting cells and T cells to augment tumor cell lysis [25,26].”

New references inserted:

[25] Lyssiotis et al 2017, 10.1016/j.tcb.2017.06.003

[26]Vecchio et al 2021,10.3389/fcell.2021.730726)

Reviewer 2 Report

The review article by Gibson et al. is very well written and organized. The authors have linked cancer metabolism and HIV infection for future research as well. The only missing part in this review is there are some other cancer-causing viruses that also encode miRNA and regulate several signaling pathways linked to metabolism.

Author Response

Reviewer #2 comments

The review article by Gibson et al. is very well written and organized. The authors have linked cancer metabolism and HIV infection for future research as well. The only missing part in this review is there are some other cancer-causing viruses that also encode miRNA and regulate several signaling pathways linked to metabolism.

Author's Reply to the Review Report (Reviewer 2)

We thank reviewer #2 for the time they took to assess our manuscript. The suggestion to integrate information on the modulation of metabolism by miRNAs encoded by tumor viruses is absolutely pertinent and we thank the reviewer for bringing this to our attention. We have introduced a new paragraph at the end of section 3.1 (bottom of page 9) that discusses this aspect:

“Interestingly, some oncogenic DNA viruses have been shown to express miRNAs that interfere with metabolic pathways. The Kaposi’s Sarcoma Herpesvirus (KSHV) and the Epstein Barr Virus (EBV) account for a considerable percentage of the DNA virus associated cancers. Although their oncogenic mechanisms have been strongly linked to the NF-kB signaling pathway [64], the modulation of cellular metabolism by viral miRNAs has been clearly shown to promote a metabolic shift akin to the Warburg effect. Kaposi’s Sarcoma Herpesvirus (KSHV) encoded miRNA expression reduces mitochondrial biogenesis and oxygen consumption while increasing lactate secretion and glucose uptake and promoting aerobic respiration through the activation of HIF-1α signaling [65]. EBV encoded microRNA EBV-miR-BART1 has been shown to be highly expressed in many EBV-associated tumors. Work by Ye and colleagues [66] has shown that this induces modifications in cellular metabolism by changing the glycolytic flux, through phosphoglycerate dehydrogenase (PHGDH), which have previously been shown to contribute to oncogenesis [67]. These examples emphasize the power of miRNAs to contribute to the metabolic reprograming required for cancer cells to proliferate efficiently.”

New references inserted:

[64] Charostad et al, doi:10.1186/s13027-020-00317-4.

[65] Yogev et al, doi:10.1371/journal.ppat.1004400.

[66] Ye et al, doi:10.1016/j.bbrc.2013.05.008.

[67] Locasale et al, doi:10.1038/ng.890.